

# Development and validation of the Health Promoting Behaviour for Bloating (HPB-Bloat) scale

Nurzulaikha Abdullah[1,*], Yee Cheng Kueh[1,*], Garry Kuan[2,3], Mung Seong Wong[4,5], Fatan Hamamah Yahaya[6], Nor Aslina Abd Samat[4,5], Khairil Khuzaini Zulkifli[5,7] and Yeong Yeh Lee[4,5]

[1] Biostatistics & Research Methodology Unit, School of Medical Sciences, Universiti Sains Malaysia, Kubang Kerian, Kelantan, Malaysia
[2] Exercise and Sport Science Programme, School of Health Sciences, Universiti Sains Malaysia, Kubang Kerian, Kelantan, Malaysia
[3] Department of Life Sciences, Brunel University, London, United Kingdom
[4] Department of Medicine, Universiti Sains Malaysia, Kubang Kerian, Kelantan, Malaysia
[5] GI Function & Motility Unit, Hospital Universiti Sains Malaysia, Universiti Sains Malaysia, Kubang Kerian, Kelantan, Malaysia
[6] School of Distance Education, Universiti Sains Malaysia, Pulau Pinang, Malaysia
[7] Faculty of Medicine, Universiti Teknologi MARA, Sungai Buloh, Selangor, Malaysia
* These authors contributed equally to this work.

Corresponding author
Yee Cheng Kueh, yckueh@usm.my

## ABSTRACT

**Background:** Health management strategies may help patients with abdominal bloating (AB), but there are currently no tools that measure behaviour and awareness. This study aimed to validate and verify the dimensionality of the newly-developed Health Promoting Behaviour for Bloating (HPB-Bloat) scale.

**Methods:** Based on previous literature, expert input, and in-depth interviews, we generated new items for the HPB-Bloat. Its content validity was assessed by experts and pre-tested across 30 individuals with AB. Construct validity and dimensionality were first determined using exploratory factor analysis (EFA) and Promax rotation analysis, and then using confirmatory factor analysis (CFA).

**Results:** During the development stage, 35 items were generated for the HPB-Bloat, and were maintained following content validity assessment and pre-testing. One hundred and fifty-two participants (mean age of 31.27 years, 68.3% female) and 323 participants (mean age of 27.69 years, 59.4% male) completed the scale for EFA and CFA, respectively. Using EFA, we identified 20 items that we divided into five factors: diet (five items), health awareness (four items), physical activity (three items), stress management (four items), and treatment (four items). The total variance explained by the EFA model was 56.7%. The Cronbach alpha values of the five factors ranged between 0.52 and 0.81. In the CFA model, one problematic latent variable (treatment) was identified and three items were removed. In the final measurement model, four factors and 17 items fit the data well based on several fit indices (root mean square error of approximation (RMSEA) = 0.044 and standardized root mean squared residual (SRMR) = 0.052). The composite reliability of all factors in the final measurement model was above 0.60, indicating acceptable construct reliability.

**Conclusion:** The newly developed HPB-Bloat scale is valid and reliable when assessing the awareness of health-promoting behaviours across patients with AB. Further validation is needed across different languages and populations.

# INTRODUCTION

Abdominal bloating (AB) is a common symptom that can be associated with impaired quality of life and psychological dysfunction. The most recent global epidemiology survey of functional gastrointestinal (GI) disorders involved 73,000 respondents and found that the prevalence of functional AB and distension was 3.5% and 1.2%, respectively (*Sperber et al., 2020*). There are a number of lifestyle risk factors that may trigger or aggravate AB, including physical inactivity, stress, and obesity (*Cai Lian et al., 2016*; *Cook & Schoeller, 2011*; *Graff et al., 2006*). However, studies on the effects of bowel disorders, such as AB, on lifestyle routines have yielded disparate findings (*Fernández-Bañares et al., 2006*; *Lacy, Weiser & Lee, 2009*).

The majority of differences in self-reported lifestyles are related to sedentary behaviours and eating habits. *Prince et al. (2008)* found that physical inactivity and sedentary behaviours were associated with fluctuating mortality and morbidity rates (*WHO, 2018*). Researchers have also found that stress, anxiety, sleep problems, and somatic symptoms were independent predictors of GI disorders (*Nicholl et al., 2008*), including bloating (*O'Malley et al., 2011*). According to the National Health and Morbidity Survey (NHMS), mental health problems were observed in 29.2% of Malaysians aged 16 and older (*Mustapha, 2018*). Other less common, but equally as important, risk factors for AB are eating and dietary habits. A university student in Taiwan reportedly died from stomach cancer after consuming instant noodles on a daily basis and having chronic bloating, nausea, and stomachache symptoms (*NST, 2018*). Eating an imbalanced diet can lead to obesity, which can cause many different health problems (*Rashid, 2017*; *WHO, 2019*). Obesity is recognized as an important contributing factor to GI symptoms including AB (*Delgado-Aros et al., 2004*; *Ho & Spiegel, 2008*).

A study by *Kua et al. (2012)* reported that 4.2% of patients in Singapore that sought self-medications were from the cohort that experienced bloating symptoms. However, positive improvements can be seen using non-drug approaches for bloating (*Khoshoo, Armstead & Landry, 2006*; *Lacy, Weiser & Lee, 2009*), including changing dietary habits (*Fernández-Bañares et al., 2006*), using pro- and/or prebiotics (*Kim et al., 2005*; *McFarland & Dublin, 2008*; *Moayyedi et al., 2010*; *Vulevic et al., 2018*), cognitive-behaviour therapy (*Boyce et al., 2000*; *Drossman et al., 2003*), herbs (*Liu et al., 2006*; *Vejdani et al., 2006*), and ointment or massage (*Lämås et al., 2009*; *Lotfipur-Rafsanjani et al., 2018*). Treatments for known disorders that cause bloating, such as constipation (*Han et al., 2018*; *Lämås et al., 2009*), can also help bloating (*Foley et al., 2014*; *Johannesson, 2015*). However,

each treatment used on its own has different reported success rates due to heterogeneity in use, dose, and compliance.

Due to the limitations described above, creating an environment and adopting good practices and behaviour in order to facilitate a healthy lifestyle may be a more suitable strategy in AB management. This is the basis for the theory of planned behaviour (TPB) (*Viner & Macfarlane, 2005*). The framework of the TPB consists of belief, intention, and behaviour, with behaviour as the central core (*Ajzen, 1991*). Self-management behaviours were observed to influence the quality of life in patients with type 2 diabetes mellitus (*Kueh et al., 2015*; *Kueh, Morris & Ismail, 2017*). The TPB proposes that beliefs (based on attitudes, subjective norms, and perceived behavioural control) influence intention, which further affects behaviour. Past behaviour can also act an additional predictor of a person's current intention and behaviour (*McEachan & Conner, 2011*; *Thomson, White & Hamilton, 2012*; *Abdullah et al., 2020*). In addition to looking at the effects of past behaviours on current intentions and behaviours, the efficacy of the TPB in predicting health-related behaviours is also influenced by behaviour type, sample characteristics, and methodological factors (*McEachan & Conner, 2011*). Therefore, exploring suitable health-promoting behaviours is important in order to increase awareness across people who suffer with AB. A validated tool that can evaluate the health-promoting behaviours of AB patients is needed, and our study aimed to develop and validate a scale that could assess these behaviours.

## MATERIALS AND METHODS

### Study design, sampling method, and participants

Using a cross-sectional study design with purposive sampling, we conducted our study between May 2018 and October 2019. A total of 520 people from the compound of the Hospital Universiti Sains Malaysia (HUSM), Kelantan, Malaysia were screened. Ultimately, 510 (98.1%) participants were deemed eligible and were recruited for the study. The participants consisted of patients, caregivers, accompanying persons, hospital staff, and students around HUSM. The inclusion criteria were: age 18 years and older, a functional bloating diagnosis (based on the Rome IV criteria or a clinical diagnosis), and/or experience of bloating at least once in the past 3 months (based on an answer to the verbal question "Have you ever experienced bloating?" and/or using a pictogram from the Rome foundation). A clinical diagnosis of bloating was made based on the experience of the physicians who managed functional GI disorders. The exclusion criteria were: an absence of a history of organic GI diseases (e.g., inflammatory bowel disease, GI infections, and colorectal cancer), a history of abdominal surgeries, taking drugs that may cause or worsen bloating (e.g., opiates), and any severe psychiatric illnesses (e.g., schizophrenia). Appropriate eligibility criteria is important and can impact external validity when designing a study (*Patino & Ferreira, 2018*). These specific criteria were chosen in order to cover the adult general population that were not affected by severe diseases.

## Ethical approval

Ethical approval was obtained from the Human Research Ethics committee, Universiti Sains Malaysia (USMKK/PPP/JEPEM/17010012) prior to the start of the study. This study also conformed to the guidelines set by the International Declaration of Helsinki. Written informed consent was obtained from each participant.

## Developing the HPB-Bloat

The new HPB-Bloat scale was developed to measure health-promoting behaviours across people with AB, and was based on the TPB, one of the most commonly-used and well-validated decision-making models that examines attitude and behaviour. We employed the theory, driven with the approach of dimension/indicator analysis that was described by Hox (1997). Based on our research of the literature and discussions with the research team experts, we conceptualized AB health behaviours across five domains: diet, health awareness, physical activity, stress management, and treatment. The new item generation was conducted by the researchers through an extensive literature review related to health behaviours that could encourage improvements in AB symptoms. Based on the literature review, a total of 24 items were generated. The research team experts also provided an additional 10 related items and supported the five temporary domains from the early draft of the HPB-Bloat scale. In order to cover all of the important indicators for the behaviour construct, we conducted an in-depth interview of 12 individuals with AB symptoms. The in-depth interview was conducted using guided questions. For example, "What do you think about AB in daily life?", and "How do you manage AB in daily life?" Additional probing questions that focused on specific activities used to manage symptoms included "How about your dietary intake? Does it contribute to improve your AB symptoms?", "How about physical activity? Does it contribute to improve your AB symptoms?", "How about stress management? Does it contribute to improve your AB symptoms?", and "Are there any other things that help you deal with AB? If so, is it helpful and how does it help?". The duration of the interview was approximately 30 min to 1 h. All the recorded interviews were transcribed into a transcript, which was then narratively analysed. Themes were identified from the transcript, a theme list was created, and interview segments were coded. Important aspects and critical points from the interviewed individuals were identified. From these interviews, we found an additional item that we added to the HPB-Bloat's item pool. Hence, a total of 35 items were generated in the initial stage of developing the first draft of the HPB-Bloat. The responses for each item were rated using a five-point Likert-scale, from never (1) to very often (5). All items were developed in the Malay language, which is the main spoken language in the study's location. The first draft of the HPB-Bloat was then examined for its content validity by seven invited experts, who each had at least 10 years of experience in the GI field, psychometric testing, language, and questionnaire development. Figure 1 shows the item generation process from the initial stage of development to the final stage of item reduction for the newly developed HPB-Bloat.

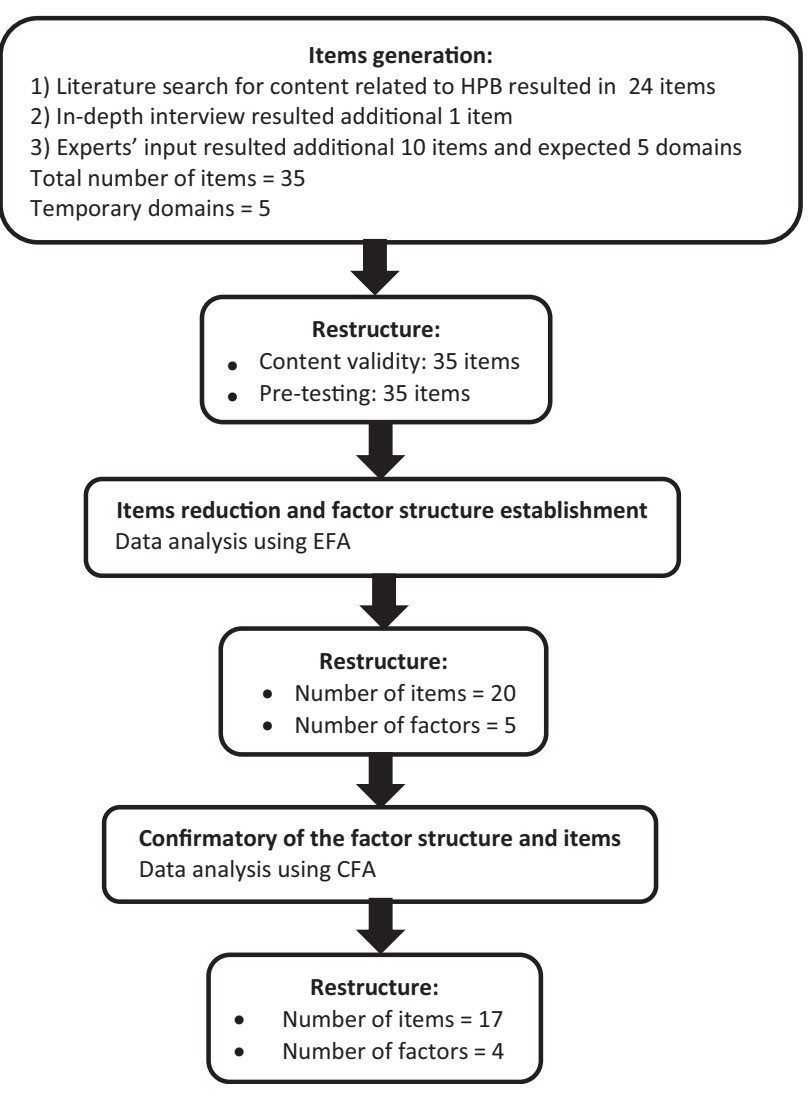

**Figure 1 Summary of questionnaire development process.**

## Content validity and pre-testing of the HPB-Bloat

To assess content validity, we computed content validity indices for the items (I-CVI) and scale (S-CVI) based on the relevant responses from the invited experts (*Lynn, 1986*; *Polit & Beck, 2006*). Using the average method (S-CVI/Ave), we found that the items' content validity index and the scale's content validity index for the five expected domains (or factors) in the HPB-Bloat were all more than 0.75, which were considered satisfactory (*Lynn, 1986*). Subsequently, we conducted pre-testing on 30 participants diagnosed with functional bloating (based on Rome IV or a clinical diagnosis). The invited participants were asked to comment on the clarity and comprehensibility of the administered HPB-Bloat. The format and font sizes were modified based on suggestions from the participants. However, the structure and wording of items remained unchanged.

Based on the results from the CVIs (using I-CVI and S-CVI/Ave) and pre-testing, the first version of the HPB-Bloat remained at 35 items. In order to determine the validity and reliability of this HPB-Bloat draft, we performed an exploratory factor analysis (EFA), followed by a confirmatory factor analysis (CFA).

## Data and statistical analysis

The EFA and CFA were performed using the Statistical Package for Social Sciences (SPSS) version 24.0 (IBM, Armonk, NY, USA) and Mplus 8 (*Muthén & Muthén, 1998*). The variables were expressed as mean and standard deviation (SD) for numerical items and frequency (n) and percentage (%) for categorical variables.

The EFA was first performed on the initial 35 items in order to explore and extract the major contributing factors, using the principal axis factoring with Promax rotation (Kappa 4). We decided the number of factors based on the eigenvalue, which should be greater than one. Factor loadings greater than 0.40 were regarded as significantly relevant and items with lower loadings were considered problematic and were deleted (*Hair et al., 2010*). All item deletions were conducted in a consecutive manner and subsequent models were re-specified following each deletion. The reliability of the factors was checked using internal consistency based on Cronbach's alpha, with the generally recommended threshold value of 0.60 (*Taber, 2018*). However, a value above 0.50 is still considered acceptable by some literature (*George & Paul Mallery, 2003*; *Hinton, Brownlow & McMurray, 2004*; *Karin, Sabine & Marcel, 2019*).

Following EFA, we tested the final structure model with CFA to further confirm its validity. Items with factor loadings less than 0.40 were removed after adequate theoretical support was carried out by researchers. The modification indices (MI) were used as a guide to improve the model by adding the items' residual correlation. Modifications to the model were based on theoretical basis and expert opinion.

The measurement CFA model was assessed using several fit indices (*Kline, 2011*). The fit indices and the recommended threshold values were: a standardized root mean squared residual (SRMR) value lower than 0.080, a root mean square error of approximation (RMSEA) value lower than 0.070, and a comparative fit index (CFI) and Tucker-Lewis Fit Index (TLI) values above 0.92 for number of items 12 to 30 (*Hair et al., 2010*). The CFA model was re-specified with adequate theoretical support until most of the model fit indices met the criteria.

We used the correlation matrices among the latent constructs to establish discriminant validity. The composite reliability of the measurement CFA model was computed (*Hair et al., 2010*). Composite reliability greater than the threshold value of 0.60 was considered acceptable (*Hair et al., 2010*).

## RESULTS

A total of 160 and 350 HPB-Bloat scales were distributed to AB patients for EFA and CFA, respectively. One hundred and fifty-two (95.0%) and 323 (92.3%) participants completed the HPB-Bloat scales for EFA and CFA, respectively. The mean age of respondents for

EFA was 31.27 years old (SD = 14.36) and 68.3% were female. For CFA, the mean age of respondents was 27.69 years old (SD = 11.50) and 59.4% were male.

## The EFA model

Our results indicated adequate sampling and reliable estimates (Kaiser-Meyer-Olkin Measure of Sampling Adequacy (KMO) = 0.732, Bartlett's test of sphericity= $p < 0.001$; *Kaiser, 1974*; *Yong & Pearce, 2013*). The 35 initial items, divided into five factors, were represented with a cumulative value of 56.7% of the variance, which indicated acceptable importance. The variance value for each of the five factors was 22.25, 10.91, 9.58, 7.26, and 6.65, respectively.

We removed the problematic items from the subsequent EFA models until we achieved a final model with all factor loadings greater than 0.40. Three items with factor loadings less than 0.40 (T5 = 0.349, SM1 = 0.287, and D2 = 0.370) were kept for further analysis. Eventually, we removed 15 items and the final model consisted of 20 items. The descriptive statistics, factor loading, and internal consistency results of the final EFA model are summarized in Table 1. The five factors described were diet, health awareness, physical activity, stress management, and treatment. The internal consistency corresponded with the reported Cronbach's alphas, ranging from a low of 0.52 (stress management) to a high of 0.81 (health awareness).

## The CFA model

As shown in Table 2, the results of the initial 20-item CFA model revealed that none of the model fit indices met the criteria. Additionally, the CFA output showed that the latent variable covariance matrix was not positive definite, and there was a problem involving the treatment factor. Therefore, we subjected the initial model to re-specification for improvement. The CFA output indicated that the treatment factor had a high standardized correlation (>1) with the diet factor. However, we found that both factors had items that made them not suitable to be combined into one factor. We further investigated the meaning of the four items under the treatment factor. We removed these items iteratively from the initial CFA model, except for item T4 ("I will always be ready to try new treatment techniques to improve AB symptoms when needed"), which has a treatment awareness component. Therefore, we grouped item T4 under the health awareness factor and renamed the factor as health and treatment awareness. After removing the treatment factor, Model-1 consisted of four factors with 17 items (Table 3). The fit indices were improved in Model-2, but CFI and TLI were still not within the acceptable threshold values. Based on the MI results and following adequate theoretical support carried out by the researchers, we added residual correlations from several items within the same factor into the model, one by one. The final model (Model 2) fit the data well based on several fit indices: CFI = 0.929, TLI = 0.911, SRMR = 0.052, and RMSEA = 0.044 (0.032, 0.061) (Table 2, Model-2).

As Table 3 illustrates, all standardized factor loadings exceeded the threshold of 0.40. The composite reliability of all Model-2 factors was greater than 0.60, which indicated good reliability.
**Table 1 Results of descriptive statistics, EFA, and reliability.** The 5 factors were described as diet, health awareness, physical activity, stress management, and treatment.

| No. abbreviated item content | Mean | SD | Factor loading | | | | |
|---|---|---|---|---|---|---|---|
| | | | 1 | 2 | 3 | 4 | 5 |
| D1 | 4.17 | 0.75 | 0.480 | | | | |
| D2 | 4.07 | 0.85 | 0.370 | | | | |
| D3 | 4.23 | 0.76 | 0.743 | | | | |
| D4 | 3.98 | 0.91 | 0.599 | | | | |
| D5 | 3.84 | 0.78 | – | | | | |
| D6 | 4.33 | 0.53 | 0.697 | | | | |
| D7 | 3.95 | 0.72 | – | | | | |
| D8 | 3.37 | 0.95 | – | | | | |
| D9 | 3.81 | 0.74 | – | | | | |
| HA1 | 3.57 | 0.83 | | 0.484 | | | |
| HA2 | 3.75 | 0.80 | | 0.755 | | | |
| HA3 | 3.89 | 0.66 | | – | | | |
| HA4 | 3.56 | 0.91 | | 0.671 | | | |
| HA5 | 3.84 | 0.74 | | 0.902 | | | |
| HA6 | 3.31 | 0.89 | | – | | | |
| HA7 | 4.04 | 0.56 | | – | | | |
| PA1 | 3.80 | 0.80 | | | – | | |
| PA2 | 3.62 | 1.09 | | | 0.521 | | |
| PA3 | 3.67 | 0.83 | | | 0.773 | | |
| PA4 | 3.67 | 0.84 | | | 0.539 | | |
| PA5 | 4.08 | 0.76 | | | – | | |
| PA6 | 3.89 | 0.90 | | | – | | |
| SM1 | 4.13 | 0.62 | | | | 0.287 | |
| SM2 | 4.12 | 0.79 | | | | 0.495 | |
| SM3 | 4.06 | 0.54 | | | | – | |
| SM4 | 3.93 | 0.58 | | | | 0.561 | |
| SM5 | 3.59 | 0.92 | | | | – | |
| SM6 | 4.02 | 0.83 | | | | – | |
| SM7 | 4.25 | 0.65 | | | | – | |
| SM8 | 3.63 | 1.05 | | | | 0.543 | |
| T1 | 3.48 | 1.17 | | | | | – |
| T2 | 4.26 | 0.52 | | | | | 0.405 |
| T3 | 3.96 | 0.61 | | | | | 0.751 |
| T4 | 3.92 | 0.66 | | | | | 0.517 |
| T5 | 3.92 | 0.60 | | | | | 0.349 |
| Eigenvalue | | | 4.45 | 2.18 | 1.92 | 1.45 | 1.33 |
| Variance explained (%) | | | 22.25 | 10.91 | 9.58 | 7.26 | 6.65 |
| Cumulative variance (%) | | | 22.25 | 33.17 | 42.75 | 50.00 | 56.66 |
| Cronbach alpha | | | 0.74 | 0.81 | 0.64 | 0.52 | 0.58 |

**Note:**
D, Diet; HA, Health awareness; PA, Physical activity; SM, Stress management; T, Treatment.

**Table 2 Summary for HPB-Bloat model fit indices.** The final model (Model 2) fit the data well based on several fit indices.

| Path model | RMSEA (90% CI) | CFI | TLI | SRMR |
|---|---|---|---|---|
| Model-0[a] | 0.062 [0.053–0.070] | 0.828 | 0.796 | 0.064 |
| Model-1[b] | 0.064 [0.054–0.075] | 0.842 | 0.809 | 0.062 |
| Model-2[c] | 0.044 [0.032–0.061] | 0.929 | 0.911 | 0.052 |

**Notes:**
[a] Model-0 with original model with 5 factors and 20 items.
[b] Model-1 with deleted problematic items; T2, T3, T5.
[c] Model-2 with additional correlated items residual; T4 with HA5, T4 with HA4, T4 with HA1, D6 with D2, SM4 with SM2.

**Table 3 Standardized factor loading (λ), and composite reliability of CFA discriminant validity among latent variables of CFA in Model 2.** All standardized factor loadings have exceeded the threshold of 0.40. All correlations between factors were below 0.85 which suggest that discriminant validity of the HPB-Bloat was satisfied.

| Constructs/items | Mean | SD | Model-0 | Model-1[#] | Model-2[#] | |
|---|---|---|---|---|---|---|
| | | | λ | λ | λ | CR |
| Diet | | | | | | 0.77 |
| D1 | 4.04 | 0.74 | 0.57 | 0.60 | 0.59 | |
| D2 | 3.91 | 0.92 | 0.51 | 0.56 | 0.61 | |
| D3 | 4.14 | 0.86 | 0.65 | 0.61 | 0.60 | |
| D4 | 3.95 | 0.91 | 0.68 | 0.72 | 0.70 | |
| D6 | 4.20 | 0.68 | 0.66 | 0.60 | 0.65 | |
| Health awareness | | | | | | 0.82 |
| HA1 | 3.85 | 0.84 | 0.54 | 0.51 | 0.55 | |
| HA2 | 4.11 | 0.70 | 0.65 | 0.62 | 0.63 | |
| HA4 | 3.98 | 0.82 | 0.71 | 0.67 | 0.72 | |
| HA5 | 3.99 | 0.76 | 0.79 | 0.83 | 0.77 | |
| Treatment | | | | | | |
| T2 | 4.31 | 0.76 | 0.30 | – | – | |
| T3 | 4.15 | 0.82 | 0.49 | – | – | |
| T4 | 4.01 | 0.76 | 0.60 | 0.67* | 0.67* | |
| T5 | 4.10 | 0.69 | 0.66 | – | – | |
| Physical activity | | | | | | 0.64 |
| PA2 | 4.13 | 0.83 | 0.48 | 0.48 | 0.46 | |
| PA3 | 4.02 | 0.80 | 0.73 | 0.73 | 0.74 | |
| PA4 | 4.03 | 0.78 | 0.62 | 0.62 | 0.62 | |
| Stress management | | | | | | 0.69 |
| SM1 | 4.40 | 0.60 | 0.45 | 0.46 | 0.43 | |
| SM2 | 4.03 | 0.82 | 0.62 | 0.63 | 0.76 | |
| SM4 | 3.97 | 0.83 | 0.60 | 0.59 | 0.73 | |
| SM8 | 3.93 | 0.96 | 0.49 | 0.49 | 0.47 | |

**Note:**
λ, standardized factor loading; CR, composite reliability, all factor loadings were statistically significant at $p < 0.050$.
* T4 was grouped into health awareness and the factor was renamed as health and treatment awareness.
[#] The four-factor model consists of latent variables diet, health and treatment awareness, physical activity, stress management.

**Table 4 Discriminant validity among latent variables of CFA in Model 2.** All correlations between factors were below 0.85 which suggest that discriminant validity of the HPB-Bloat was satisfied.

| Constructs/Correlation coefficient, r | 1 | 2 | 3 | 4 |
|---|---|---|---|---|
| 1. Diet | 1 | 0.83 | 0.68 | 0.51 |
| 2. Health and treatment awareness | | 1 | 0.58 | 0.59 |
| 3. Physical activity | | | 1 | 0.66 |
| 4. Stress management | | | | 1 |

**Note:**
All correlation coefficients were statistically significant at $p < 0.001$.

As shown in Table 4, all correlations between factors were below 0.85, which suggested that the HPB-Bloat's discriminant validity was satisfactory.

## DISCUSSION

The development of the HPB-Bloat and the evaluation of its validity and reliability are vital steps in assessing health-promoting behaviours among Malay-speaking patients who suffer from AB symptoms. The newly developed HPB-Bloat, based on the concept of TPB, has been proven to meet the validity and reliability standards through a multi-phase approach. The final version of the 17-item HPB-Bloat with four factors is ready to be used in future studies to evaluate health-promoting behaviours across the Malaysian population experiencing AB symptoms.

There are a few scales that measure health-promoting behaviours, for instance, the Health Promoting Lifestyle Profile (HPLP) (*Duffy, Rossow & Hernandez, 1996*; *Paudel et al., 2017*; *Walker, Sechrist & Pender, 1987*), the HPLP-II (*Malakouti et al., 2015*; *Mirghafourvand et al., 2015*; *Wei et al., 2012*), the Adolescent Health Promotion Scale (AHPS) (*Ortabag et al., 2011*), and the Wellbeing and Health Promotion survey (*El Ansari et al., 2011*; *El Ansari & Stock, 2010*). The HPLP and HPLP-II were developed to prevent diseases and lessen morbidity, while subsequently improving quality of life and cutting healthcare costs (*Kuan et al., 2019*; *Lim et al., 2016*; *Mirghafourvand et al., 2015*). The HPB-Bloat was introduced for these same reasons, but specifically for people with AB.

Few studies have explored the connection between health-promoting behaviour and other causal factors such as social support, physical activity, gender, family size, obesity, and well-being (*Baheiraei et al., 2011*; *Hubbard, Muhlenkamp & Brown, 1984*; *Mirghafourvand et al., 2015*; *Wainwright, Thomas & Jones, 2000*). *Sousa et al. (2015)* suggested that a person's lifestyle factors accounted for 60% of their quality of health and life. The HPLP-II questionnaire is a scale that is commonly used to measure a person's overall health-promoting behaviours and lifestyle. However, AB patients often use different self-management strategies, which encouraged us to develop a new health-promoting behaviour scale specifically for AB.

In stressful or fast-paced environments, people often adopt unhealthy lifestyles and can develop chronic illnesses, including AB. Lifestyle management through health-promoting behaviour is a possible solution to this problem, and has been shown to improve disease and quality of life (*Kuan et al., 2019*; *Musavian et al., 2014*). The newly developed

HPB-Bloat scale may assist various health stakeholders, including physicians, psychologists, public health professionals, and patients themselves, in evaluating health-promoting behaviours related to AB self-management. We recommend that future studies apply this new scale in different populations in order to analyze the stability of its performance.

There were strengths, but also limitations to the study. First, this study included individuals from the community (hospital compound) that had experienced AB, rather than solely hospital-based patients. This was done so that the scale could be used to evaluate the general public rather than only patients. Second, we purposively sampled only from the northeastern region of Peninsular Malaysia, and therefore our results may not be generalizable to other populations. Additionally, the scale was designed to be applicable to the adult population and cannot be used to assess bloating or health-promoting behaviours in adolescents or children. Finally, the original scale was developed in the Malay language, so further validation studies with the scale translated into other languages, such as English, are needed.

## CONCLUSION

In conclusion, we conducted several series of validation process to confirm that the newly developed HPB-Bloat scale and its four factors have good construct validity and structure. Future studies should apply the new HPB-Bloat across different populations, languages, and different health stakeholders in order to test its validity and stability over time.

## ACKNOWLEDGEMENTS

We would like to thank all of the study's participants and supporting members.

### Funding

This work was supported by the Research University Individual Grant from Universiti Sains Malaysia (1001.PPSP.8012250). The funders had no role in study design, data collection and analysis, decision to publish, or preparation of the manuscript.

### Grant Disclosures

The following grant information was disclosed by the authors:
Universiti Sains Malaysia: 1001.PPSP.8012250.

### Competing Interests

Yeong Yeh Lee is an Academic Editor for PeerJ.

### Author Contributions

- Nurzulaikha Abdullah conceived and designed the experiments, performed the experiments, analyzed the data, prepared figures and/or tables, authored or reviewed drafts of the paper, involved in questionnaire development, and approved the final draft.

- Yee Cheng Kueh conceived and designed the experiments, performed the experiments, analyzed the data, prepared figures and/or tables, authored or reviewed drafts of the paper, involved in questionnaire development, and approved the final draft.
- Garry Kuan conceived and designed the experiments, performed the experiments, authored or reviewed drafts of the paper, involved in questionnaire development, and approved the final draft.
- Mung Seong Wong performed the experiments, authored or reviewed drafts of the paper, involved in questionnaire development, and approved the final draft.
- Fatan Hamamah Yahaya performed the experiments, authored or reviewed drafts of the paper, involved in questionnaire development, and approved the final draft.
- Nor Aslina Abd Samat performed the experiments, authored or reviewed drafts of the paper, involved in questionnaire development, and approved the final draft.
- Khairil Khuzaini Zulkifli performed the experiments, authored or reviewed drafts of the paper, involved in questionnaire development, and approved the final draft.
- Yeong Yeh Lee conceived and designed the experiments, performed the experiments, authored or reviewed drafts of the paper, involved in questionnaire development, and approved the final draft.

### Human Ethics
The following information was supplied relating to ethical approvals (i.e., approving body and any reference numbers):

Ethical approval was obtained from the Human Research Ethics committee, Universiti Sains Malaysia (USMKK/PPP/JEPEM/17010012).

### Data Availability
The raw data are available in the Supplementary Files.

### Supplemental Information
Supplemental information for this article can be found online at http://dx.doi.org/10.7717/peerj.11444#supplemental-information.

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
