# Peer review of "Development and validation of the Health Promoting Behaviour for Bloating (HPB-Bloat) scale"

_PeerJ, doi:10.7717/peerj.11444_

## Round 0.1 · original submission · Major Revisions

English editing is mandatory.
The manuscript has improved, however, it is necessary to make a further revision in order to satisfy the indications of the Reviewers.

Reviewer 1 ·

Basic reporting

Clear and unambiguous, professional English used throughout.
In some places, for example page 2, lines 72-73, references are required.
Authors are advised to use "scale" word everywhere for designed tool; in some places in the article, the word questionnaire is used.
Isn't it better to cite to the reference of "Mplus 8", statistical analysis software for the structural equations modeling in the study?

Experimental design

Methods were not described with sufficient detail & information to replicate:
1- Design of study is sequential exploratory mixed method and it is necessary for the author to explain the study design, sampling method and data collection & analysis plan separately in two phases (qualitative-quantitative).
2- Please correct "Hospital" in this phrase “hospital Universiti Sains Malaysia (HUSM)” and explain about setting of study.
3- Please clarify on what logic these inclusion and exclusion criteria were chosen? For example; age, past history of GI surgery, and etc. It is very important that investigators not only define the appropriate inclusion and exclusion criteria when designing a study but also evaluate how those decisions will impact the external validity of the results of the study (Patino & Ferreira, 2018).
Reference: Patino CM, Ferreira JC. Inclusion and exclusion criteria in research studies: definitions and why they matter. Jornal Brasileiro de Pneumologia. 2018 Apr;44(2):84-.
4- Research strategies and methods for conceptualization and Operationalization must were clearly explained. The theoretical concepts (conceptualization) and its translation into measurable variables (operationalization) need specification. According to Hox and De Jong-Gierveld (1990), there are two perspectives to bridge the gap between theory (concept) and observable variables (operationalization or measurement): i) the theory driven approaches (as dimension/indicator analysis, semantic analysis and facet design methods) and ii) the empirical driven approaches (as content sampling, symbolic interactionism and concept mapping), It is not clear which approach was used by the authors of the article. Please explain this part for the article readers.
References:
Hox, J and De Jong-Gierveld, J.J., eds. (1990). Operationalization and Research Strategy, Lisse (NL): Swets & Zeitlinger
Hox, J.(1997). From Theoretical Concept to Survey Question. In Survey Measurement and Process Quality, eds L. Lyberg, P.P. Biemer, M. Collins, E. de Leeuw, C. Dippo, N. Schwarz, and D.Trewin. New York: John Wiley and Sons.
5- It is better to transfer the part in which authors of manuscript, explain the theory of TPB (P. 4, lines 139-146) to the introduction of the article because in the introduction, this theory is discussed as a basis for making decisions in adopting healthy lifestyle behaviors.
6- Was the in-depth interview with the 12 participants qualitatively analyzed? How? In what way? Were the dimensions of behavior, including diet, health awareness, physical activity, stress management and treatment explored by analysis of interviews or by reviewing literature related to the concept you intended to design a scale to measure? It is necessary to explain this part in detail and how to extract the items of initial scale and how the item pool was generated. How many items were obtained from reviewing literature and how many items were obtained from analyzing interviews? How did you make sure the content domain was well developed?
7- Wouldn't it be better to ask the behaviors frequency mentioned in the items, or to ask how often these behaviors were performed instead of severity agreement with the item? Because more in attitude measuring tools, the individual's agreement with the items is questioned. This can lead to bias and respondents expressing their opinions instead of expressing their behavior.
8- How was the content validity index for the scale calculated? Was the average method (SCVI-Ave) or the universal agreement method (SCVI-UA) used? Please clarify.
9- Was the content validity ratio (CVR) not determined for the scale items?

Validity of the findings

1- Statistical analyzes and the findings are well explained in detail. Just please clarify about deleted items. Mention these items in detail. It is not clear what attributes measured by the deleted items. Didn't removing them from the scale damage the comprehensiveness of the scale’s items to measure the concept of the bloating health promoting behavior? How was this ensured?
2- I suggest that authors add a graph to the article to illustrate the process of item reduction in the research stages and specify on what basis the items were reduced at each stage.

Additional comments

The second paragraph of the discussion is a kind of repetition of the results of factor analysis and needs to be adjusted. Please in limitation of study, also note that the designed scale is applicable to the adult population and cannot be used to assess bloating-promoting behaviors in adolescents and children.

Annotated reviews are not available for download in order to protect the identity of reviewers who chose to remain anonymous.

Reviewer 2 ·

Basic reporting

The English language is good throughout the manuscript. However, some grammatic errors need to be revised:

Line 52: Cronbach alpha values…ranged ‘from’ 0.52 to 0.81
Line 57: Composite reliability of all factors…above 0.60, which indicated…
Line 59: Further validation…delete ‘also’
Line 82: Reword the sentence to ‘mental health problem was observed in 29.2% Malaysians aged 16 and above’
Line 83: risk factor’s’ for AB are eating or dietary habit
Line 120: Delete 'However'
Line 182: ...considered problematic ‘and’ would be deleted
Line 230: change ‘subject for’ to ‘subject to’

Experimental design

Line 147: Please provide the reason for selecting these domains
Line 150: Please provide detailed information about the ‘in-depth’ interview (did you use any questionnaires? What are the questions?...)

Validity of the findings

Line 211: The test for sampling adequacy should be Kaiser-Meyer-Olkin test, not Kaplan Maier Olkin test. KMO values between 0.8 and 1 indicate adequate sampling. Since your KMO is 0.732, this is not a meritorious value. Please indicate this point in the discussion section

Additional comments

The authors designed a tool to assess the awareness of behavior among patients with abdominal bloating. They demonstrated the reliability and validity of this tool. However, the authors need to provide more detailed information in the experimental design section.

---

## Round 0.2 · Minor Revisions

English editing is required since there are some errors.

Further, the reviewer gives some minor criticisms that need to be taken into account.

Reviewer 1 ·

Basic reporting

no comment

Experimental design

Please add the interview questions, average time of interviews, and the method for analyzing of the interviews data after line 176.
Please add the method using for reliability in line 208.
Lines 370-373 in discussion are duplication the results of study and can be removed.

Validity of the findings

no comment

Additional comments

Thank you for the corrections made.
Looks like your article needs to be edited by a native English speaker. There are some mistakes. For example "From these interviews" instead of "this interview", because you have 12 participants. In the line 184: "Responses of each items were rated in five-point...", instead of "Responses of each items were measured in five-point". "Eigen" and "techniques" instead of "eigen" and "technics", respectively.
"To assess content validity, the content validity indexes for items (I-CVI) and scale (S-CVI) were computed based on the responses of relevancy from invited experts (Lynn, 1986; Polit & Beck, 2006). The results of item’s content validity index and scale’s content validity index based on average method (S-CVI/Ave), for the five expected domains (or factors) in HPB-Bloat were all more than 0.75, a finding which was considered satisfactory (Lynn, 1986)." instead of "To assess content validity, the content validation indexes (CVI) for items and scale were computed based on responses of relevancy from invited experts (Lynn, 1986; Polit & Beck, 2006). The results of item’s content validity (I-CVI) and scale’s content validity ,(S-CVI) for the five expected domains (or factors) in HPB-Bloat were all more than 0.75, a finding which was considered satisfactory (Lynn, 1986)." and others.

---

## Round 0.3 · accepted · Accept

Dear Dr. Kueh,

I would like to thank you very much for your submission to PeerJ.
Congratulations!

Yours,

Yoshi

Prof. Yoshinori Marunaka, M.D., Ph.D.
Academic Editor, PeerJ